

# Circular RNA hsa_circ_0051246 acts as a microRNA-375 sponge to promote the progression of gastric cancer stem cells *via* YAP1

Minghui Deng[1,2], Yefeng Xu[2], Yongwei Yao[2], Yiqing Wang[2], Qingying Yan[2], Miao Cheng[2] and YunXia Liu[2]

[1] Department of Oncology, The First Affiliated Hospital of Anhui Medical University, Hefei, Anhui, China
[2] Department of Oncology, Hangzhou Third People's Hospital, Hangzhou, Zhejiang, China

## ABSTRACT

**Background**. Gastric cancer (GC) stem cells play an important role in GC progression. Circular RNAs (circRNAs) act as microRNA (miRNA) sponges and inhibit the biological function of miRNAs in GC cytoplasm. MiRNAs also participate in GC progress. circ_0051246 was shown to be associated with miR-375 after analyzing GC microarray data GSE78091 and GSE83521. The oncoprotein Yes-associated protein 1 (YAP1) is targeted by miR-375 and can be inactivated *via* the Hippo tumor suppressor pathway. Due to insufficient research on circ_0051246, this study aimed to investigate its relationship with miR-375 and YAP1 in cancer stem cells (CSCs).

**Methods**. SGC-7901 CSCs were used to establish knockdown/overexpression models of circ_0051246, miR-375, and YAP1. Malignant phenotypes of CSCs were assessed using Cell Counting Kit 8, colony/sphere formation, 5-Ethynyl-2′-deoxyuridine assay, flow cytometry, Transwell, and wound healing assays. To detect the interactions between circ_0051246, miR-375, and YAP1 in CSCs, a dual-luciferase reporter assay and fluorescence *in situ* hybridization were performed. In addition, 24 BALB/c nude mice were used to establish orthotopic xenograft tumor models. Four groups of mice were injected with CSCs ($1 \times 10^6$ cells/100 µL) with circ_0051246 knockdown, miR-375 overexpression, or their respective control cells, and tumor progression and gene expression were observed by hematoxylin-eosin staining, immunohistochemistry. Western blot and quantitative real-time PCR were utilized to examine protein and gene expression, respectively.

**Results**. Circ_0051246 silencing reduced viability, promoted apoptosis, and inhibited proliferation, migration and invasion of CSCs. The functional effects of miR-375 mimics were comparable to those of circ_0051246 knockdown; however, the opposite was observed after miR-375 inhibitors treatment of CSCs. Furthermore, circ_0051246-overexpression antagonized the miR-375 mimics' effects on CSCs. Additionally, YAP1 overexpression promoted CSC features, such as self-renewal, migration, and invasion, inhibited apoptosis and E-cadherin levels, and upregulated the expression of N-cadherin, vimentin, YAP1, neurogenic locus notch homolog protein 1, and jagged canonical notch ligand 1. Conversely, YAP1-silenced produced the opposite effect. Moreover, miR-375 treatment antagonized the malignant effects of YAP1 overexpression in CSCs. Importantly, circ_0051246 knockdown and miR-375 activation suppressed CSC tumorigenicity *in vivo*.

Corresponding author
YunXia Liu, zjhzsylyx@163.com

**Conclusion**. This study highlights the promotion of circ_0051246-miR-375-YAP1 axis activation in GC progression and provides a scientific basis for research on the molecular mechanism of CSCs.

# INTRODUCTION

Gastric cancer (GC) is an extremely common tumor of the digestive system, and its incidence and mortality rates are respectively the 5th and 4th highest in the global malignant tumors (*Siegel et al., 2022*). Although it is the leading cause of cancer-related deaths worldwide, most cases have been reported in Asian countries (*Sung et al., 2021*). Due to the atypical early symptoms and lack of screening in patients with GC, most patients are already in advanced stages at the time of diagnosis. Over the past decades, great progress has been made in the clinical diagnosis and treatment strategy for GC, but the 5-year survival rate for GC in most Asian countries is still below 30% (*Allemani et al., 2018*). One study reported that recurrence and metastasis after treatment are the main reasons for the poor prognosis of GC (*Joshi & Badgwell, 2021*). Cancer stem cells (CSCs) are small tumor cells with strong self-renewal ability and nondirectional differentiation abilities (*Batlle & Clevers, 2017*), which play a critical role in occurrence, development, recurrence, and metastasis (*Tan et al., 2020*). Evidence suggests that CSCs enriched with AQP5 are involved in the early formation of invasive GCs (*Tan et al., 2020*) indicating that they promote GC invasion. By applying magnetic strain-activated cell sorting (MACS) technology, scientists have used CD44+ cells to separate CSCs from human GC lines (*Takaishi et al., 2009*). Epithelial cell adhesion molecule (EpCAM) is also being viewed as a biomarker for GC (*Alshaer et al., 2017*), along with CD44. Both of these markers have established correlations with the malignant phenotype of GC cells (*Dai et al., 2017*; *Hou et al., 2022*). Therefore, it is crucial to regulate the malignant phenotype of CSCs in GC to improve the survival and prognosis of GC patients.

Circular RNA (circRNA) is a non-coding RNA that is produced by the back-splicing of precursor mRNA from exons in eukaryotes (*Li, Yang & Chen, 2018*). Studies have shown that circRNA expression disorders can be observed in both common malignant tumors and rare malignant tumors (*Kristensen et al., 2022*). CircRNAs can sponge microRNAs (miRNAs) in the cytoplasm, thereby, inhibiting the biological functions of miRNAs (*Hansen et al., 2013*). MiRNAs have been identified as biomarkers that play a role in malignant development of GC (*Gillen, 2021*; *Zheng et al., 2017*) with miR-375 inhibiting GC stemness (*Ni et al., 2021*). *Luo et al. (2020)* reported that sponging circ_CCDC9 inhibits miR-6792-3p expression by antagonizing the proliferation, migration, and invasion of GC. Yu with colleagues found that circ_0021087 sponges miR-184, decreasing tumorigenesis in GC cells (*Yu et al., 2021*). Similarly, another study confirmed that circ_REPS2 can

sponge miR-558 to inhibit GC progression by regulating RUNX3 (*Guo et al., 2020*). Thus, exploring circRNA expression in CSCs may aid in the early screening and treatment of GC.

To explore the potential regulatory effects of circRNA and its targeted miRNA in GC, we analyzed the GC-related circRNA expression profile GSE83521 and miRNA expression profile GSE78091 from the Gene Expression Omnibus (GEO), as previously reported by our team (*Liu et al., 2020*). According to the criteria of $|\text{Log}_2 \text{fold change(FC)}| \geq 2.0$ and $P$-value $< 0.05$, we identified 26 differentially expressed (DE) circRNAs (Fig. 1A) and 151 DE miRNAs. To predict the 196 targeted miRNAs from the 26 DE circRNAs (Fig. 1B), we used a Venn diagram to compare the 196 targeted miRNAs and 151 DE miRNAs, and identified 10 key miRNAs (Fig. 1C): hsa-miR-621, hsa-miR-648, hsa-miR-421, hsa-miR-1289, hsa-miR-1263, hsa-miR-375, hsa-miR-140-3p, hsa-miR-567, hsa-miR-1197 and hsa-miR-1205 (Fig. 1B). As a single-stranded and small non-coding RNA (*Zhang et al., 2019*), miRNAs are multifunctional and participate in the progression of various tumors (*Ciszkowicz et al., 2020*; *Kang et al., 2018*; *Wang et al., 2016*). For example, miR-375 targets several important oncogenes such as Yes-associated protein 1 (YAP1) (*Wang et al., 2016*; *Wei et al., 2021*). miR-375-3p can suppress tumorigenesis in colorectal cancer cells by inhibiting YAP1 (*Xu et al., 2019*). YAP1 is a well-known oncoprotein in GC that can be inactivated *via* the Hippo tumor suppressor pathway (*Liu et al., 2022*). According to the microarray data analysis of GSE78091 and GSE83521, circ_0051246 is related to miR-375 (*Liu et al., 2020*). Hsa-miR-375 was downregulated in GC in the miRNA expression profile of GSE78091 (Fig. 1D), whereas YAP1 was significantly upregulated (Fig. 1E).

We speculate that circ_0051246 sponges miR-375 to promote the progression of GC *via* YAP1. By establishing SGC-7901 CSC models with circ_0051246, miR-375, and YAP1 knockdown or overexpression, and an orthotopic xenograft tumor model, we provide a new target for GC screening and treatment.

## MATERIAL AND METHODS

### Microarray data analysis

As presented in our previous research (*Liu et al., 2020*), to explore circRNAs with potential regulatory effects, we screened and downloaded GC-related microarray data from the GEO database (http://ncbi.nlm.nih.gov/geo/), obtaining the circRNA expression microarray GSE83521 and the miRNA expression microarray GSE78091. Screening conditions were $|\log_2(\text{FC})| \geq 2.0$ and $P$-value $< 0.05$. We identified 26 DE circRNAs and 151 DE miRNAs from the GEO database using the interactive online tool GEO2R. Subsequently, we used the circular RNA interactome website (https://circinteractome.nia.nih.gov/index.html) (*Dudekula et al., 2016*) to predict 196 miRNAs that may interact with 26 DE circRNAs. A Venn diagram (*via* OmicShare tools, https://www.omicshare.com/tools/Home/Soft/venn) was used to obtain the intersection of 151 DE miRNAs (from GSE78091) and 196 miRNAs (from predictions), and ten miRNAs were identified. Additionally, we used gene expression profiling interactive analysis (GEPIA) (*Tang et al., 2017*) to determine YAP1 expression in stomach adenocarcinoma.

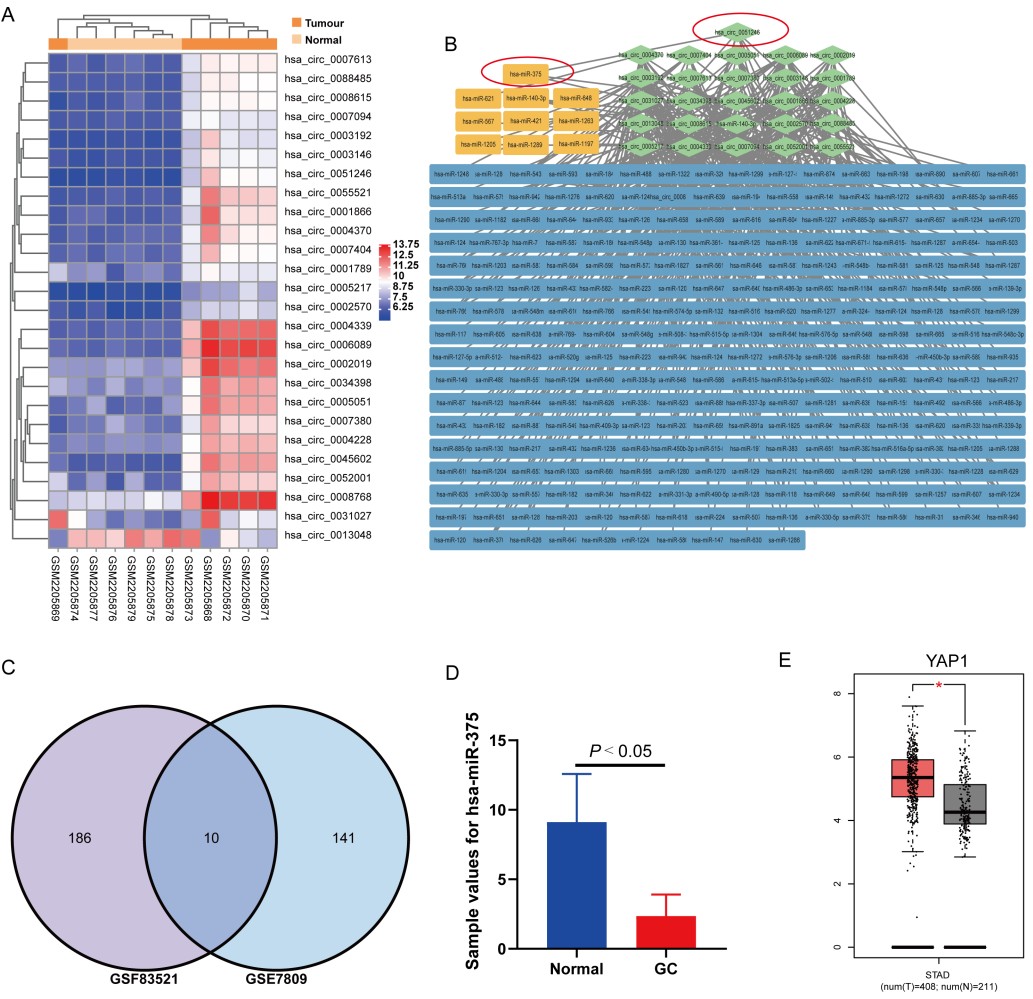

**Figure 1** **Analysis results of circRNA expression microarray GSE83521 and the miRNA expression microarray GSE78091.** (A) Heatmap of 26 differentially expressed (DE) circRNAs with |Log$_2$fold change| ≥ 2.0 and $P$-value < 0.05 in microarray GSE83521. (B) Network diagram of 26 DE circRNAs (green) and their predicted miRNAs (blue and yellow), totaling 196. The related genes are connected by gray lines. The yellow predicted miRNAs are indicated by the intersection in the (C) Venn diagram. Venn diagram shows 196 predicted miRNAs from GSE8352, 151 DE miRNAs from GSE78091, and the ten miRNAs from the previous 196 targeted miRNAs and 151 DE miRNAs. The ten miRNAs are hsa-miR-621, hsa-miR-648, hsa-miR-421, hsa-miR-1289, hsa-miR-1263, hsa-miR-375, hsa-miR-140-3p, hsa-miR-567, hsa-miR-1197 and hsa-miR-1205. (D) The miR-375 expression in human GC and adjacent normal gastric mucosa tissues (from the miRNA expression microarray GSE78091). (E) There were the YAP1 expression in human stomach adenocarcinoma and normal samples from genotype tissue expression data and The Cancer Genome Atlas Program normal data. *$P$ < 0.05 $vs$. normal sample group.

## Cell culture

SGC-7901 (C6795, Beyotime, China) and AGS (iCell-h016, iCell Bioscience, China) cell lines (human gastric adenocarcinoma cells), and the GES-1 (GES) cell line (iCell-h062, iCell Bioscience, China) (human gastric mucosal epithelial cells) were cultured in RPMI 1640 (R20161, Yuanye, China) containing 10% fetal bovine serum (FBS; 10099141C, Gibco, USA) in 10% CO$_2$ at 37 °C.

## CSCs sorting and flow cytometry

SGC-7901 CSCs were obtained using immunomagnetic bead sorting (*Dalerba et al., 2007*). Human CD44 (20 µL; 130-095-194) and CD326 (EpCAM) (20 µL; 130-061-101) microbeads were added into 80 µL cell suspension ($1 \times 10^7$ cells) and incubated in the dark for 15 min at 4 °C. Microbeads were purchased from Miltenyi Biotec (Bergisch Gladbach, Germany). Cells were then centrifuged ($300 \times g$, 10 min) and resuspended using autoMACS running buffer (130-091-221; Miltenyi Biotec) to separate the CD44$^+$ EpCAM$^+$ SGC-7901 cells using a MidiMACS separator (Miltenyi Biotec, Bergisch Gladbach, Germany).

Flow cytometry was used to detect the expression of cell biomarkers. Phenol red and EDTA-free trypsin (15090046; Gibco, USA) were used to digest the cells. The centrifuged cells were resuspended and incubated with antibodies against CS44-FITC (ab30405; Abcam, Cambridge, UK) and EpCAM-PE (ab239311; Abcam, Cambridge, UK) antibodies in the dark and on ice for 30 min. The results were measured and analyzed using a flow cytometer (C6; BD Biosciences) and FlowJo10.0 software (BD Biosciences, Franklin Lakes, NJ, USA). Additionally, apoptosis was measured by flow cytometry using $1 \times 10^6$ cells/mL cells and the Annexin V-FITC kit (556547; BD Biosciences) as described by *Bieszczad et al. (2020)*.

## Cell transfection and grouping

Lentiviruses carrying short hairpin RNA (shRNA) were used to target circ_0051246 (sh-circ_0051246 #1, #2, and #3) and targeted YAP1 (sh-YAP1) and puromycin (1 µg/mL; ST551; Beyotime, Jiangsu, China) was administered for three weeks to establish knockdown cells. Additionally, plasmids carrying circ_0051246, YAP1 cDNA, hsa-miR-375-5p mimics, inhibitors, and agomir were used with Lipofectamine 3000 (Invitrogen, Waltham, MA, USA) for 48 h to establish cells with circ_0051246 and YAP1 overexpression, miR-375 activation, or suppression. The lentivirus, plasmid and their empty vectors were provided by Jikai Gene Chemical Technology Co., Ltd. (Shanghai, China).

First, SGC-7901 CSCs were divided into four groups, namely sh-NC, sh-circ_0051246, circ-NC, and circ_0051246, to explore the effects of circ_0051246 on CSCs in GC. Next, to explore the effects of miR-375 on GC and SGC-7901 GSCs, GSCs were divided into control, miR-375 mimic, miR-375 inhibitor, and circ_0051246 + miR-375 mimic groups. To explore the effects of YAP1 and the effect of miR-375 on YAP1, the CSCs were divided into the control, the YAP1, the sh-YAP1, and the miR-375 mimic + YAP1 groups. Finally, the SGC-7901 CSCs were divided into sh-NC, sh-circ_0051246, agomiR-NC, and agomiR-375 groups to establish a xenograft model.

## Quantitative real-time PCR (qRT-PCR)

The supernatant from the lysed cell suspension was used to isolate RNA precipitates using chloroform and isopropanol successively. Subsequently, the precipitates were rinsed with 75% absolute ethanol. The precipitation was dissolved in 20 µL DEPC water (R0601; Thermo Fisher Scientific, Waltham, MA, USA) and standby. MiScript II RT Kit (218161; Qiagen, Hilden, Germany) and HiFiScript cDNA Synthesis Kit (CW2569; Cwbio, Beijing, China) were used for miRNA and mRNA reverse transcription, respectively. Finally, the SYBR Premix Ex Taq II (RR3090R; Takara, Shiga, Japan) was used for PCR

**Table 1  The information of primer sequences.**

| Gene | Forward Primer (5′–3′) | Reverse Primer (5′–3′) |
|---|---|---|
| Human miR-375 | AAGCTTGGCTGATGCTGAGAAG | TCTAGACGGCCCCGGGTCTTC |
| Human U6 | AAAGCAAATCATCGGACGACC | GTACAACACATTGTTTCCTCGGA |
| Human YAP1 | TAGCCCTGCGTAGCCAGTTA | TCATGCTTAGTCCACTGTCTGT |
| Human GAPDH | ACAACTTTGGTATCGTGGAAGG | GCCATCACGCCACAGTTTC |
| Human circ0051246 | ATTGAACGGGTGCCTAGAGAAG | ACGGCGCAGAACAGAAAACG |
| Human GAPDH | ACAACTTTGGTATCGTGGAAGG | GCCATCACGCCACAGTTTC |

amplification. *GAPDH* and *U6* were used as internal references. All qRT-PCR experiments were independently repeated thrice. The relative quantitative method ($2-^{\Delta\Delta}Ct$) was used to analyze the data. Table 1 shows information regarding primers used.

## Immunofluorescence (IF) staining

IF assays were performed on CSCs in 24-well plates, as previously described (*Tang et al., 2020a*). After 24 h, the cells were sequentially treated with 4% formaldehyde (1004965000, Sigma, Japan), 5% bovine serum albumin (ST023; Beyotime, China), and 0.5% Triton X-100 (X100; Sigma-Aldrich, St. Louis, MO, USA). Then, cells were incubated with anti-CD44 (ab254530, Abcam, UK) and anti-EpCAM antibodies (#46403; Cell Signaling, Danvers, MA, USA) at 4 °C overnight. The next day, cells were incubated with anti-mouse IgG H&L (ab150113; Abcam) antibody at 25 °C for 1 h. Finally, after incubation with DAPI (1:1000, BS097; Biosharp; Heifei city, China), the cells were observed under an inverted fluorescence inverted microscope (Ts2-FC, Nikon, Japan).

## Cell counting kit-8 (CCK-8) assay

CCK-8 kits (HY-K0301; MedChemExpress, Monmouth Junction, NJ, USA) were used to measure cell viability in 96-well plates. After the cells were processed according to the experimental requirements, 5000 cells/well were incubated with CCK-8 reagent (10 µL/well) for 1 h. Optical density (OD) was measured using a microplate reader (450 nm) (CMaxPlus; Molecular Devices, San Jose, CA, USA). The value of the relative experimental OD/ relative control OD was used as the evaluation indicator. The relative OD value was calculated as the OD value minus that of the blank control.

## Tumor spheroid experiment

Single-cell suspensions (500 cells) were cultured for seven days using 6-well plates with low-adhesion and the five mL of FBS-free culture medium. Tumor spheres were photographed under a microscope (Olympus, Tokyo, Japan) to count tumor sphere numbers.

## Cell colony formation

CSCs (1000 cells) were treated according to grouping for 48 h and incubated in per-well of the 6-well plates with RPMI 1640 culture medium (30% FBS) for ten days. The cell culture medium was replaced with 4% paraformaldehyde to fix cells at 4 °C for 60 min. The cells were then washed and incubated with crystal violet (HY-B324A, MCE, USA) for 2 min.

## 5-Ethynyl-2′-deoxyuridine (EdU) assay

CSCs at 90% confluence ($1 \times 10^6$ cells/well) were cultured using 12-well plates. The cells were processed using the EdU cell proliferation kit (BeyoClick™, C0078s; Beyotime), observed, and photographed using an inverted fluorescence microscope.

## Transwell assay

Matrigel was used to observe invasion, was diluted at a ratio of 3:1 in serum-free culture medium and was applied to a Transwell chamber (3422; Coring, Corning, NY, USA). The cell suspension (200 μL; $2 \times 10^5$ cells) was placed in the upper chamber. After 24 h, the fixed cells were stained with a Crystal Violet solution (E607309; Sangon Biotech, Shanghai, China). Finally, the number of deep purple cells that invaded the matrigel was used as an evaluation indicator under a microscope. The migration assay was similar to the invasion assay except that Matrigel was not added.

## Wound healing assays

In 6-well plates, CSCs ($1 \times 10^5$ cells/mL) were cultured in 2% FBS until they reached 90% confluence. A 10 μL pipette tip was used to scratch cells in each well. After washing, the cells were cultured in serum-free medium. The scratch spacing was photographed under a microscope with the same eyepiece and objective lens at different time points (0, 24 and 48 h). Axio Vision (version 4.8; Zeiss, San Diego, CA, USA) was used to evaluate the migratory ability of CSCs. Data were analyzed in triplicate.

## Dual-luciferase reporter assay

According to a previous study (*Dai et al., 2021*), Lipofectamine 3000 was used to transfect wild-type (WT) and mutant (Mut) recombinant plasmids (SV40-FIREFLY_Luciferase-MCS; Promega, Madison, WI, USA). SGC-7901 CSCs were divided into four groups, including circ_0051246 WT + miR-NC/miR-375 and circ_0051246 Mut + miR-NC/miR-375 groups, to explore the interaction between miR-375 and circ_0051246. After lysis, the cells were collected and processed using a dual-luciferase reporter gene assay kit (RG027, Beyotime, China). A microplate reader was used to measure the ratio of firefly luciferase activity to *Renilla* luciferase activity.

## Fluorescence *in situ* hybridization (FISH)

Cy3-labeled probes for circ_0051246 and FAM-labeled probes for miR-375 were synthesized (Geneseed, Guangzhou, China). A FISH probe in a hybridization buffer (GenePharma, Shanghai, China) was used to incubate the CSCs for 16 h. Cell nuclei were stained with DAPI, and images were captured using a fluorescence microscope.

## Orthotopic xenograft tumor model

The BALB/c male nude mice (5 weeks, $19 \pm 2$ g) (Beijing Vital River Laboratory Animal Technology Co., Ltd.) were used to establish the orthotopic xenograft tumor model. Mice were raised under pathogen-free conditions at 22–25 °C and kept on a 12-hour light-dark cycle. Mice had free access to standard laboratory feed and water. After one-week of adaptive feeding, the nude mice were anesthetized with isoflurane (4% for induction and 2% for maintenance). Transfected CSCs with green fluorescent protein ($1 \times 10^6$

cells/100 µL) were subcutaneously injected into the serosal layer of the greater curvature side of the stomach of the corresponding nude mice (*Yin et al., 2019*). According to the 3R principle, mice ($n = 6$ per group) were divided into sh-NC, sh-circ_0051246, AgomiR-NC, and AgomiR-375 groups using a random table method. Mice were euthanized using CO2 if they experience weight loss exceeding 20%, had tumors with diameters larger than 5% of their body weight, or if the tumor ruptured. Since none of these criteria were met during the experiment, no mice were euthanized in advance. D-luciferin was administered intraperitoneally at 28 days after tumor cell inoculation in mice. The mice were anesthetized with isoflurane and placed on an intravital fluorescence imaging system for small animals (AniView 100; BLT, Guangzhou, China) to capture the tumor situation and the luminescence information. All mice were euthanized using $CO_2$ (100% $CO_2$ at a filling rate of 30% per min). Tumors were removed, fixed, and embedded in paraffin. The mice exhibited no evident discomfort; therefore, supplementary analgesia was not required. No animals were excluded, and all animal experiments were performed by professionals blinded to the group assignment, according to the Guidelines for the Humane Treatment of Laboratory Animals. All animal experiments were performed in accordance with the Animal Experimentation Ethics Committee of Zhejiang Eyong Pharmaceutical Research and Development Center (ZJEY-20220301-02).

## Hematoxylin-eosin (HE) staining and immunohistochemistry

Tumor tissue paraffin blocks were cut into 4 µm tumor paraffin slices for HE staining using an HE kit (ab245880; Abcam, Cambridge, UK). The sections were then dehydrated, made transparent, sealed, and observed under a microscope. The degree of inflammatory cell infiltration and the proportion of metastatic tumor cells are tumor histological evaluation indicators (*Kajioka et al., 2021*; *Xiao et al., 2021*). The higher the score, the more severe was the degree of infiltration and metastasis. After inactivation and antigen repair, the blocked-dewaxed slices were incubated with anti-Ki67 (1:200, ab16667; Abcam) and anti-YAP1 (1:500, ab205270, Abcam) antibodies at 37 °C for 2 h. Thereafter, the samples were treated with Rabbit IgG H&L antibody (1:5000, ab97080; Abcam) and incubated with hematoxylin (Bry-0001-01/03/04; Runnerbio, China) for 4 min at 25 °C. Sections were observed using a microscope (Eclipse E100; Nikon, Tokyo, Japan).

## Western blot (WB)

The total protein from each group was diluted to the same concentration. Proteins were denatured, electrophoresed, and then transferred to polyvinylidene fluoride membranes (1620177; Bio-Rad, Hercules, CA, USA). Blots were blocked with 5% BSA and washed. The membrane was then incubated with the following antibodies overnight. The antibodies were anti-YAP1 (1:1000, ab205270; Abcam), anti-neurogenic locus notch homolog protein 1 (Notch 1) (1:1000, ab52627; Abcam), anti-jagged canonical notch ligand 1 (Jagged 1) (1:1000, ab7771, Abcam), anti-GAPDH (1:10000, AF7021; Affinity, Shanghai, China), anti-Ki67 (1:1000, ab16667, Abcam), anti-PCNA (1:2000, ab92552; Abcam, UK), anti-Bcl-2-associated X protein (Bax) (1:5000, ab32503; Abcam), anti-Bcl-2 (1:5000, ab32124, Abcam), anti-N-cadherin (1:5000, ab245117; Abcam), anti-Vimentin (1:1000; Abcam), and
anti-E-cadherin (1:2000, ab40772; Abcam) antibodies. After washing, the membranes were incubated with anti-rabbit (7074, CST, USA) or anti-mouse (7076, CST) IgG HRP-linked antibodies. Proteins were visualized using a chemiluminescence imaging analysis system (610020-9Q; Qinxiang, China) and ImageJ software (NIH, Bethesda, MD, USA).

## Statistical analysis

All statistical analyses were performed using SPSS Statistics (version 25.0.; IBM, USA) and Graphpad Prism 9.0 (GraphPad Software, La Jolla, CA, USA) was used to present the statistical analysis results. Multigroup data that were normally distributed and conformed to the homogeneity of variance tests were analyzed using the one-way analysis of variance (ANOVA) followed by the Turkey honest significant difference test, and two groups of data were analyzed using the $t$-test. The data were analyzed using a non-parametric Kruskal-Wallis rank sum test in cases where a normal distribution was not observed and the variance was not homogeneous. All data are expressed as the mean $\pm$ standard deviation (SD), $P < 0.05$ was used as the threshold for significant differences.

# RESULTS

## Circ_0051246 in SGC-7901 CSCs with a high expression level

Based on the analysis of GSE83521 and GSE78091 (Figs. 1A–1D), we have found the potential correlation between miR-375 and GCs, and circ_0051246 was predicted to be related to miR-375 (*Liu et al., 2020*). To the best of our knowledge, studies on the effects of circ_0051246 on CSCs derived from GCs are lacking. Therefore, we performed qRT-PCR to observe circ_0051246 expression levels and found that it had a higher expression level in SGC-7901 and AGS cells compared to human gastric mucosal epithelial cells (GES group) ($P < 0.01$) (Fig. 2A). After separating and collecting SGC-7901 CSCs (CD44$^+$ EpCAM$^+$ cells) using flow cytometry, we found that circ_0051246 was overexpressed in CSCs ($P < 0.05$) (Fig. 2B). The cell microstructure, actin in the cytoskeleton, and biomarkers of SGC-7901 CSCs were identified (Figs. 2C–2E). The cell structure was clear and actin was densely distributed in the cytoplasm. Additionally, CD44 and EpCAM expression were also observed on the cell membrane (Fig. 2E). We selected SGC-7901 CSCs for subsequent experiments.

## Circ_0051246 promotes the progression of SGC-7901 CSCs

To observe the effects of circ_0051246 on SGC-7901 CSCs, we used the cells with sh-circ_0051246 #1, #2, #3, or circ_0051246 treatment. Circ_0051246 expression in SGC-7901 CSCs was reduced to 47% after treatment with sh-circ_0051246 #3 compared to that in the sh-NC group. Conversely, overexpression of circ_0051246 in CSCs led to a 72% increase in its expression ($P < 0.01$) (Fig. 3A). Accordingly, the sh-circ 0051246 #3 was used to establish circ_0051246 knockdown cells. We observed the proliferation and self-renewal of CSCs using CCK-8 assays, sphere formation, colony formation assays, and EdU cell proliferation imaging analysis. We found that circ_0051246 knockdown inhibited cell viability, the number of spheroids, colony number, and the proportion of EdU-positive cells (Figs. 3B, 3D–3F) ($P < 0.05$), whereas circ_0051246 overexpression increased the above levels in CSCs cells ($P < 0.01$) (Figs. 3C–3E and 3G).

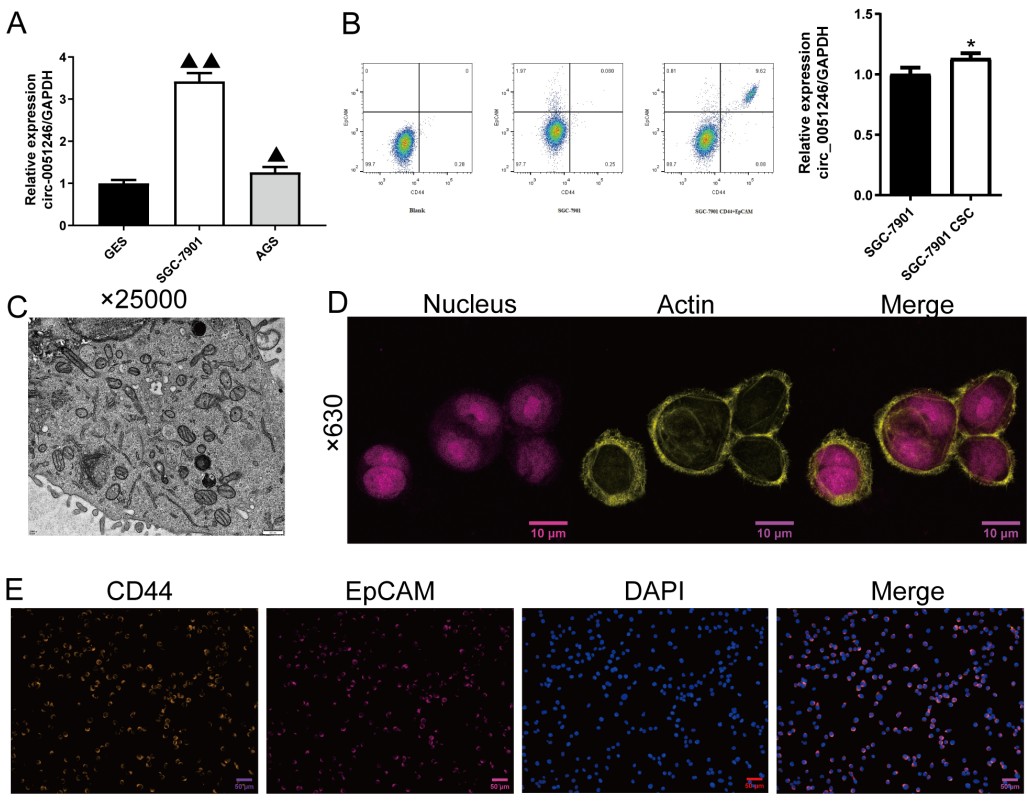

**Figure 2** **Screening and identification of SGC-7901 cancer stem cells (*n* = 3, mean ± SD).** (A) The circ_0051246 levels in human gastric mucosal epithelial cell line GES-1 (GES) and gastric cancer cell lines (SGC-7901, AGS) were measured using quantitative real-time PCR. ▲▲*P* < 0.01 *vs.* GES group. (B) Screening of cancer stem cells (CSCs) in SGC-7901 gastric cell lines using flow cytometry; circ_0051246 levels in SGC-7901 cells and CSCs were measured using quantitative real-time PCR. (C) Morphology of the CSCs observed using electron microscope (×25,000, scale bar = 500 nm). (D) Observation of actin in the cytoskeleton of CSCs using laser confocal microscopy. The nucleus is indicated in purple and actin in yellow (×630, scale bar = 10 μm). (E) Expression levels of CD44 and EpCAM in the CSCs were observed by immunofluorescence (×630, scale bar = 50 mm). CD44 is indicated in green, EpCAM in red, and DAPI in blue. *P* < 0.05 *vs.* SGC-7901 group.

Furthermore, we examined the role of circ_0051246 in CSC apoptosis using flow cytometry. Silencing circ_0051246 increased the apoptosis level and overexpression of circ_0051246 decreased apoptosis compared to their control cells (*P* < 0.05, *P* < 0.01) (Fig. 4A). Transwell and wound-healing assays were used to detect cell migration and invasion abilities. Circ_0051246 knockdown decreased migrated and invaded cell numbers and inhibited the migration distance percentage at 24 h and 48 h (*P* < 0.01) (Figs. 4B, 4D–4E). In contrast, circ_0051246 overexpression increased these indicators (*P* < 0.05) (Figs. 4C–4E).

Western blot was used to measure the proliferation-related proteins Ki67 and PCNA, the apoptosis-related Bax and Bcl-2, the migration and invasion-related proteins N-cadherin, Vimentin, and E-cadherin, and the YAP1, Notch 1, and Jagged 1 proteins levels (Figs. 4F–4H). The oncoprotein YAP1 is targeted by miR-375 and can be inactivated *via* the

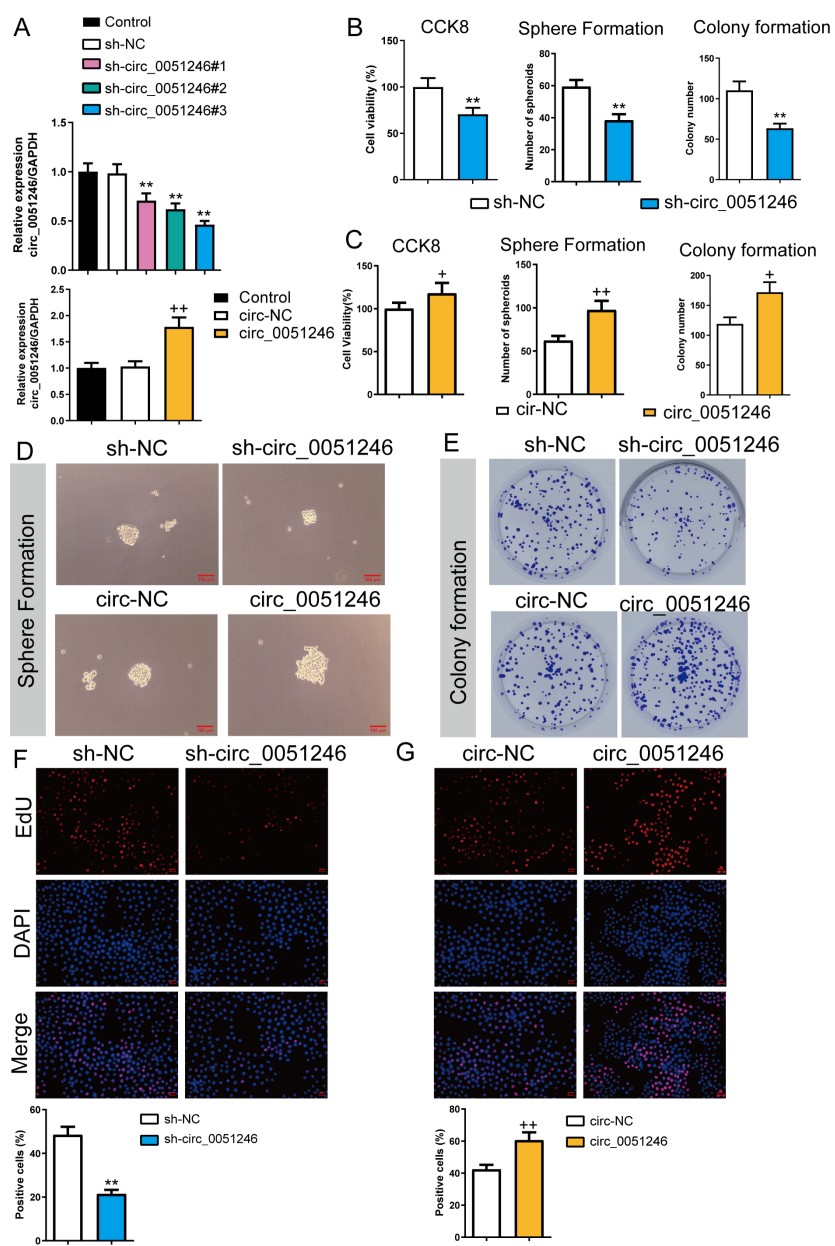

**Figure 3** **Circ_0051246 promoted the proliferation and self-renewal in SGC-7901 cancer stem cells (CSCs) ($n = 3$, mean ± SD).** (A) The circ_0051246 levels in the CSCs with knockout and overexpression of circ_0051246 were measured using quantitative real-time PCR. (B) The effects of silenced circ_0051246 on the viability, self-renewal ability, and colony-forming ability in CSCs were observed using the CCK-8 assay, sphere formation, and colony formation assay. (C) The effects of circ_0051246 overexpression on the viability, self-renewal ability, and colony-forming ability in CSCs were observed using the CCK-8 assay, sphere formation, and colony formation assay. (D) Representative photos of the sphere formation assay performed on CSCs with knockout and overexpression (scale bar = 100 μm). (E) Representative photos of the clone formation assay performed on CSCs with circ_0051246 knockout and overexpression. (F, G) Proliferation of the CSCs with circ_0051246 knockout and overexpression was measured by EdU cell proliferation imaging analysis (scale bar = 40 mm). $^{**}P < 0.01$ *vs.* sh-NC group; $^{+}P < 0.05$, $^{++}P < 0.01$ *vs.* circ-NC group.

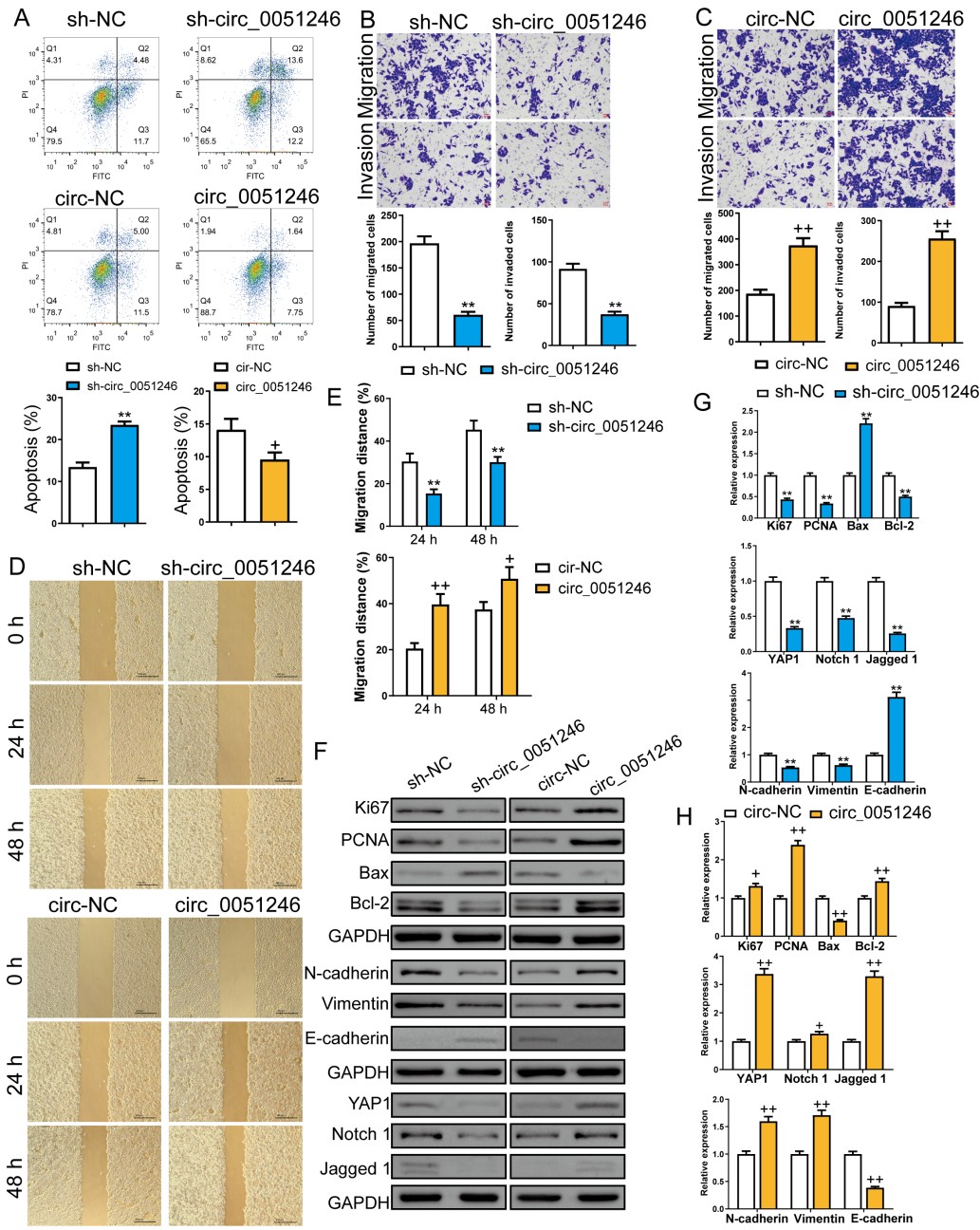

**Figure 4** **Circ_0051246 inhibited apoptosis and promoted cell migration and invasion in SGC-7901 cancer stem cells (CSCs) ($n = 3$, mean ± SD).** (A) The effects of circ_0051246 knockdown and overexpression on apoptosis in CSCs were detected using flow cytometry. (B, C) The effects of circ_0051246 knockdown and overexpression on migration and invasion in the CSCs were observed using the Transwell assay (scale bar = 40 μm). (D, E) The effects of circ_0051246 knockdown and overexpression on migration of the CSCs were observed using wound-healing assay (scale bar = 400 μm). (F) Typical protein bands indicating proteins associated with proliferation, apoptosis, and migration from Western blot. (G, H) The effects of circ_0051246 knockdown and overexpression on expression levels of proteins associated with proliferation, apoptosis, migration, as well as the YAP1 signaling pathway were measured using Western blot. $^{**}P < 0.01$ *vs.* sh-NC group; $^{+}P$ & $\lambda\tau$; 0.05, $^{++}P < 0.01$ *vs.* circ-NC group, mean ± SD.

Hippo tumor suppressor pathway (*Liu et al., 2022*; *Xu et al., 2019*). Compared to their control cells, silencing circ_0051246 decreased Ki67, PCNA, Bcl-2, YAP1, Notch1, Jagged 1, N-cadherin, and Vimentin expression levels, but silenced circ_0051246 raised the Bax and E-cadherin levels ($P < 0.05$) (Fig. 4G). Conversely, circ_0051246 overexpression increased Ki67, PCNA, Bcl-2, YAP1, Notch1, Jagged 1, and Vimentin expression levels and inhibited the Bax and E-cadherin levels ($P < 0.05$) (Fig. 4H). This suggests that circ_0051246 can regulate proliferation, apoptosis, migration, invasion, and the YAP1-related signaling pathway.

### Circ_0051246 inhibited miR-375 in SGC-7901 CSCs

To explore the interaction between circ_0051246 and miR-375, we observed the co-localization of circ_0051246 and miR-375 using FISH (Fig. 5A). The interaction between circ_0051246 and miR-375 was assessed using a dual-luciferase reporter assay. The relative fluorescence intensity was lower in circ_0051246 cells treated with miR-375 treatment ($P < 0.01$) (Fig. 5B), suggesting that circ_0051246 act as a sponge for miR-375. Furthermore, the miR-375 levels were increased in circ_0051246 knockdown CSCs, whereas miR-375 levels were decreased in CSCs overexpressing circ_0051246 ($P < 0.01$) (Fig. 5C). Circ_0051246 treatment inhibited the miR-375 levels in miR-375 mimics-treated CSCs ($P < 0.05$) (Fig. 5D).

To determine whether circ_0051246 inhibits miR-375 to regulate the malignant phenotype of SGC-7901 CSCs, we performed CCK-8, sphere formation, colony formation assays, and EdU cell proliferation imaging analysis. Treatment with miR-375 mimics inhibited cell viability, number of spheroids, colony number, and EdU-positive cell proportion ($P < 0.05$) (Figs. 5E–5H), while miR-375 inhibitors increased these levels in CSCs ($P < 0.01$) (Figs. 5E–5H). MiR-375 mimics increased CSC apoptosis, whereas miR-375 inhibitors inhibited it ($P < 0.05$) (Fig. 5I). Circ_0051246 antagonized the effects of miR-375 mimics on the proliferation, self-renewal, and apoptosis of CSCs (Figs. 5E–5I).

In addition, treatment of miR-375 mimics on SGC-7901 CSCs decreased the number of migrated and invaded cells and decreased the migration distance at 24 h and 48 h ($P < 0.05$) (Figs. 6A–6B). In contrast, treatment with the miR-375 inhibitors increased these indicators ($P < 0.01$) (Figs. 6A–6B). Moreover, miR-375 mimics decreased the Ki67, PCNA, Bcl-2, N-cadherin, Vimentin, YAP1, Notch1, and Jagged 1 expression levels and increased Bax and E-cadherin protein levels ($P < 0.05$) (Figs. 6C–6F). MiR-375 inhibitors induced opposing effects on the abovementioned proteins in SGC-7901 CSCs were completely opposite ($P < 0.05$) (Figs. 6C–6F). Moreover, circ_0051246 antagonized the effects of miR-375 mimics on the migration and invasion of SGC-7901 CSCs ($P < 0.05$) (Figs. 6A–6B). In addition, circ_0051246 antagonized the regulation of miR-375 mimics on the expression levels of proteins associated with proliferation, apoptosis, migration, invasion, and the YAP1-related signaling pathway, suggesting that circ_0051246 acts as a sponge for miR-375 to promote the malignant phenotype of SGC-7901 CSCs.

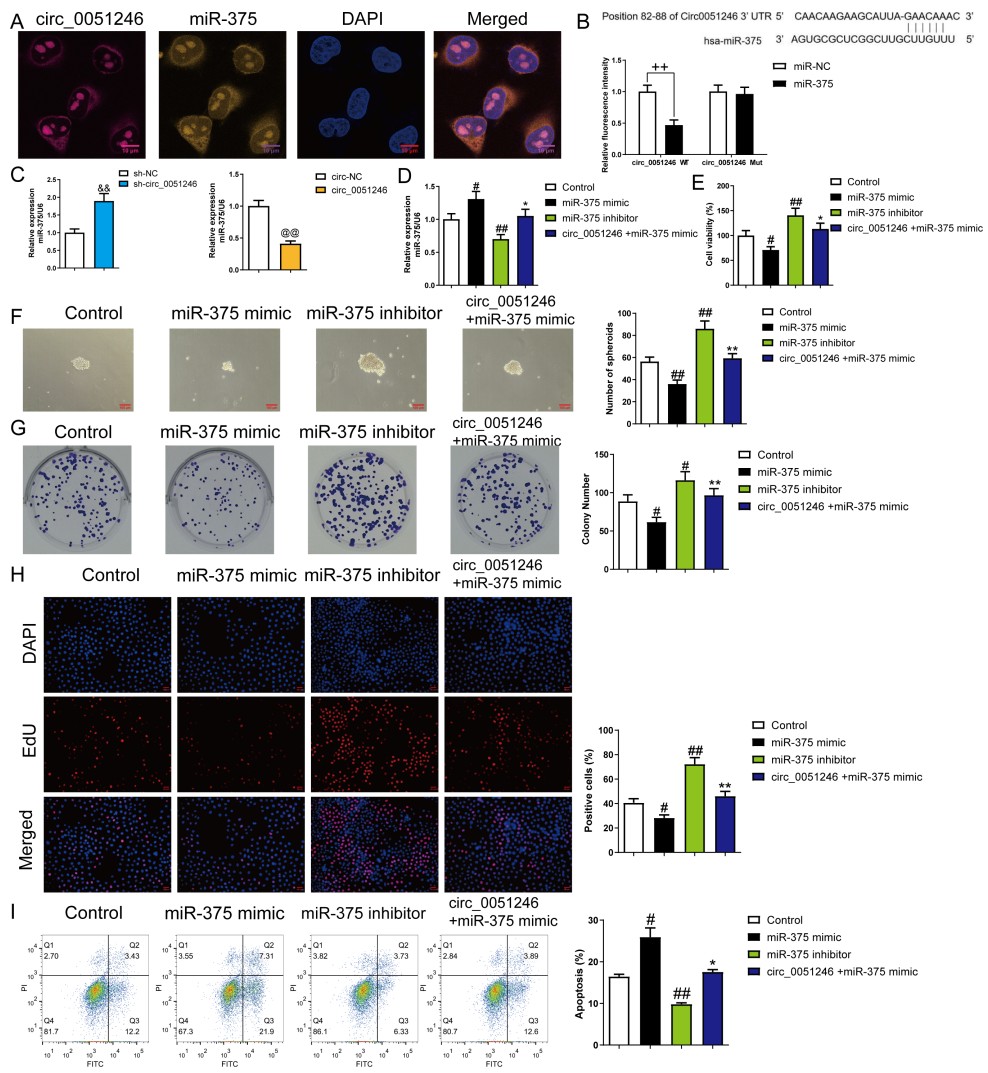

**Figure 5** **The circ_0051246 inhibited the effects of miR-375 on the proliferation, self-renewal, and apoptosis in SGC-7901 cancer stem cells (CSCs) ($n = 3$, mean ± SD).** (A) Expression levels of circ_0051246 and miR-375 in gastric CSCs were observed by using fluorescence *in situ* hybridization (scale bar = 10 μm). Circ_0051246 is indicated in purple, miR-375 in yellow, and DAPI in blue. (B) The interaction between circ_0051246 and miR-375 was verified using the dual-luciferase report assay. [++]$P < 0.01$ *vs.* miR-NC group. (C) The miR-375 level in the CSCs with circ_0051246 knockdown or overexpression were measured using quantitative real-time PCR. [&&]$P < 0.01$ *vs.* sh-NC group; [@@]$P < 0.01$ *vs.* circ-NC group. (D) The effects of circ_0051246 on the miR-375 levels in the CSCs with knockout and overexpression of miR-375 were measured using quantitative real-time PCR. (E) The effects of circ_0051246 on CSC viability with knockout and overexpression of miR-375 were measured using the CCK-8 assay. (F) The effects of circ_0051246 on self-renewal ability in CSCs with knockout and overexpression of miR-375 were observed using sphere formation assay (scale bar = 100 mm). (G) The effects of circ_0051246 on colony-forming ability in CSCs with knockout and overexpression of miR-375 were observed using the colony formation assay. (H) The effects of circ_0051246 on the proliferation of the CSCs with miR-375 knockout and overexpression were measured using EdU cell proliferation imaging analysis (scale bar = 40 μm). (I) The effects of circ_0051246 on apoptosis in the CSCs with miR-375 knockout and overexpression were detected using flow cytometry. [#]$P < 0.05$, [##]$P < 0.01$ *vs.* Control group; [*]$P < 0.05$, [**]$P < 0.01$ *vs.* miR-375 mimic group, mean ± SD.

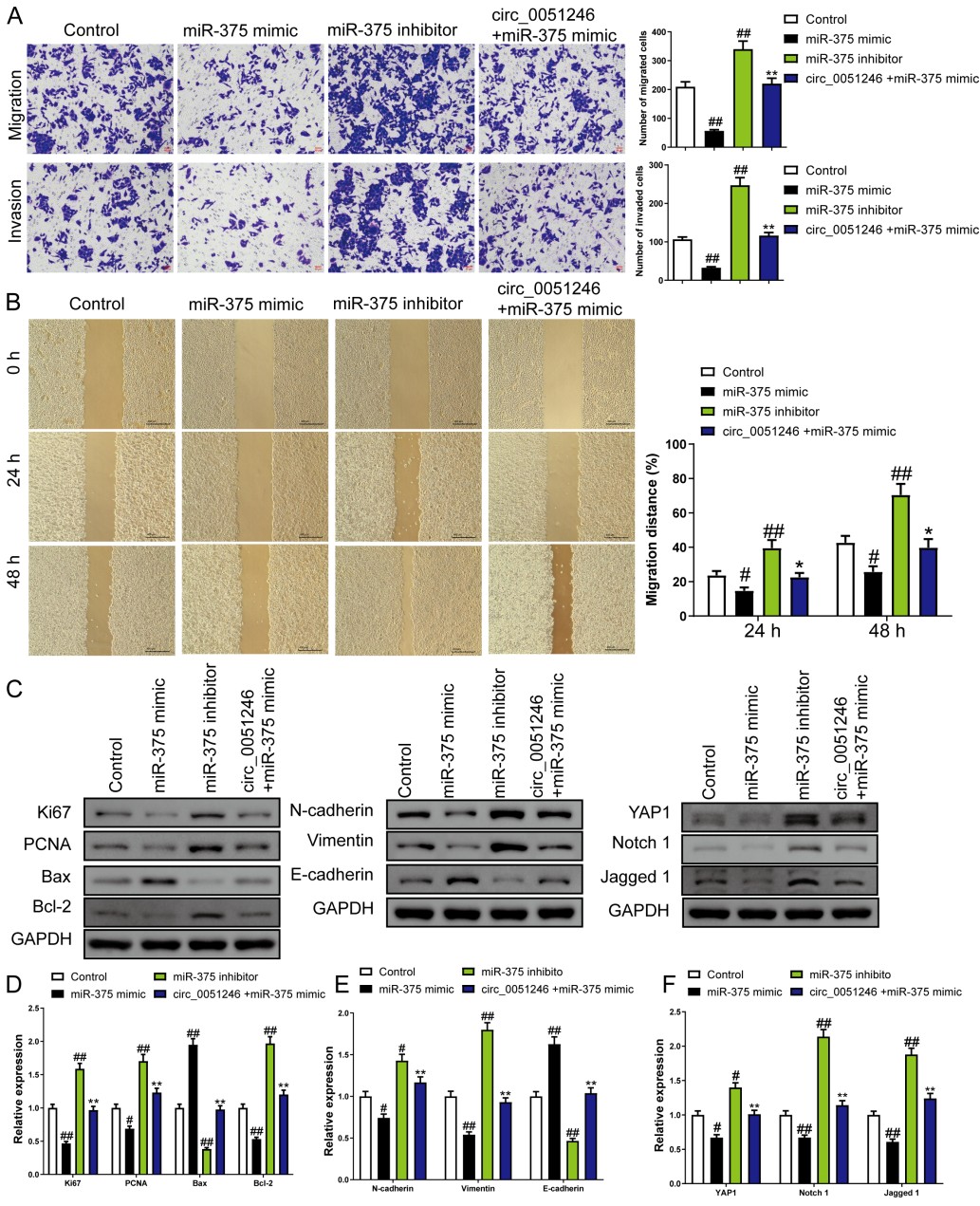

**Figure 6** **The circ_0051246 inhibited the effects of miR-375 on cell migration, invasion, and related proteins in gastric cancer stem cells (CSCs) ($n = 3$, mean ± SD).** (A) The effects of circ_0051246 on migration and invasion in the CSCs with miR-375 overexpression were observed using the Transwell assay (scale bar = 40 μm). (B) The effects of circ_0051246 on migration in CSCs with miR-375 overexpression were observed using the wound-healing assay (scale bar = 400 μm). (C) Western blot results indicating proteins related to proliferation, apoptosis, and migration, as well as YAP1 signaling pathway. The effects of circ_0051246 on expression levels of proteins related to (D) proliferation, apoptosis, (E) migration, and (F) YAP1 signaling pathway-related proteins in the CSCs with miR-375 overexpression were measured using Western blot. $^{\#}P < 0.05$, $^{\#\#}P < 0.01$ *vs.* Control group; $^{*}P < 0.05$, $^{**}P < 0.01$ *vs.* miR-375 mimic group.

## MiR-375 suppresses the progression of SGC-7901 CSCs by inhibiting YAP1

We observed that circ_0051246 overexpression increased *YAP1* mRNA level in SGC-7901 CSCs ($P < 0.05$), while circ_0051246 knockdown decreased *YAP1* ($P < 0.01$) (Fig. 7A). In addition, miR-375 mimics inhibited *YAP1* mRNA levels, whereas miR-375 inhibitors markedly increased *YAP1* mRNA level ($P < 0.05$) (Fig. 7B). Circ_0051246 overexpression antagonized the inhibition of miR-375 mimics on YAP1 expression ($P < 0.05$) (Fig. 7B). We hypothesized that miR-375 acts directly on YAP1 expression. Therefore, we used the YAP1 overexpression and knockdown in SGC-7901 CSCs.

MiR-375 mimics inhibited *YAP1* mRNA level in YAP1-overexpresed CSCs (Fig. 7C). A dual-luciferase reporter assay was used to observe the interaction between miR-375 and YAP1. SGC-7901 CSCs harboring a mutant *YAP1* gene (YAP1 Mut) were used. The relative fluorescence intensity decreased in the YAP1 WT group with miR-375 mimic treatment ($P < 0.01$) (Fig. 7D).

The proliferation and self-renewal levels of YAP1-silenced and -overexpressed CSCs were measured using the CCK-8 assay, sphere formation, colony formation assay, and EdU cell proliferation imaging analysis. YAP1 overexpression increased cell viability, spheroid number, colony number, and EdU-positive cell proportion in CSCs ($P < 0.05$) (Figs. 6F–6L), while YAP1 silencing decreased the above levels ($P < 0.05$) (Figs. 6F–6L). YAP1 overexpression led to a decrease in CSC apoptosis, while YAP1 silencing increased it ($P < 0.01$) (Fig. 6M). Furthermore, miR-375 mimics antagonized the effects of YAP1 overexpression in CSCs ($P < 0.05$) (Figs. 6F–6M).

Overexpression of YAP1 heightened the number of migrated and invaded cells and led to a higher percentage of migration at both 24 h and 48 h ($P < 0.05$) (Figs. 8A and 8B). In contrast, YAP1 silencing decreased the levels of these indicators ($P < 0.01$) (Figs. 8A–8B). Moreover, YAP1 overexpression increased N-cadherin, vimentin, YAP1, Notch1, and Jagged 1 expression levels in CSCs and inhibited the E-cadherin expression ($P < 0.05$), while YAP1 silencing induced the opposite effect ($P < 0.05$) (Figs. 8C and 8D). Moreover, miR-375 mimics treatment antagonized the effects of YAP1 overexpression on SGC-7901 CSCs ($P < 0.05$) (Figs. 8A–8D).

## Circ_0051246 knockdown and miR-375 activation inhibits the tumorigenicity of CSCs *in vivo*

An *in vivo* imaging system was used to observe the fluorescence signal intensity in an orthotopic xenograft tumor model with circ_0051246 knockdown or miR-375 activation. Circ_0051246-silenced and miR-375-activated cells inhibited the average signal strength of tumor fluorescence ($P < 0.05$) (Fig. 9A). Moreover, circ_0051246-silenced and miR-375-activated tumors had more pathological damage and inflammatory cell infiltration, and they had higher HE scores ($P < 0.05$) (Fig. 9B). In addition, Ki67 and YAP1 expression levels were observed using immunohistochemistry in orthotopic xenograft tumor mice. Circ_0051246-silenced and miR-375-activated cells showed decreased levels of Ki67 and YAP1 in the tumors ($P < 0.01$) (Figs. 9C and 9D). Circ_0051246-silenced tumors displayed lower circ_0051246 levels ($P < 0.01$), but miR-375-activated tumors did not show

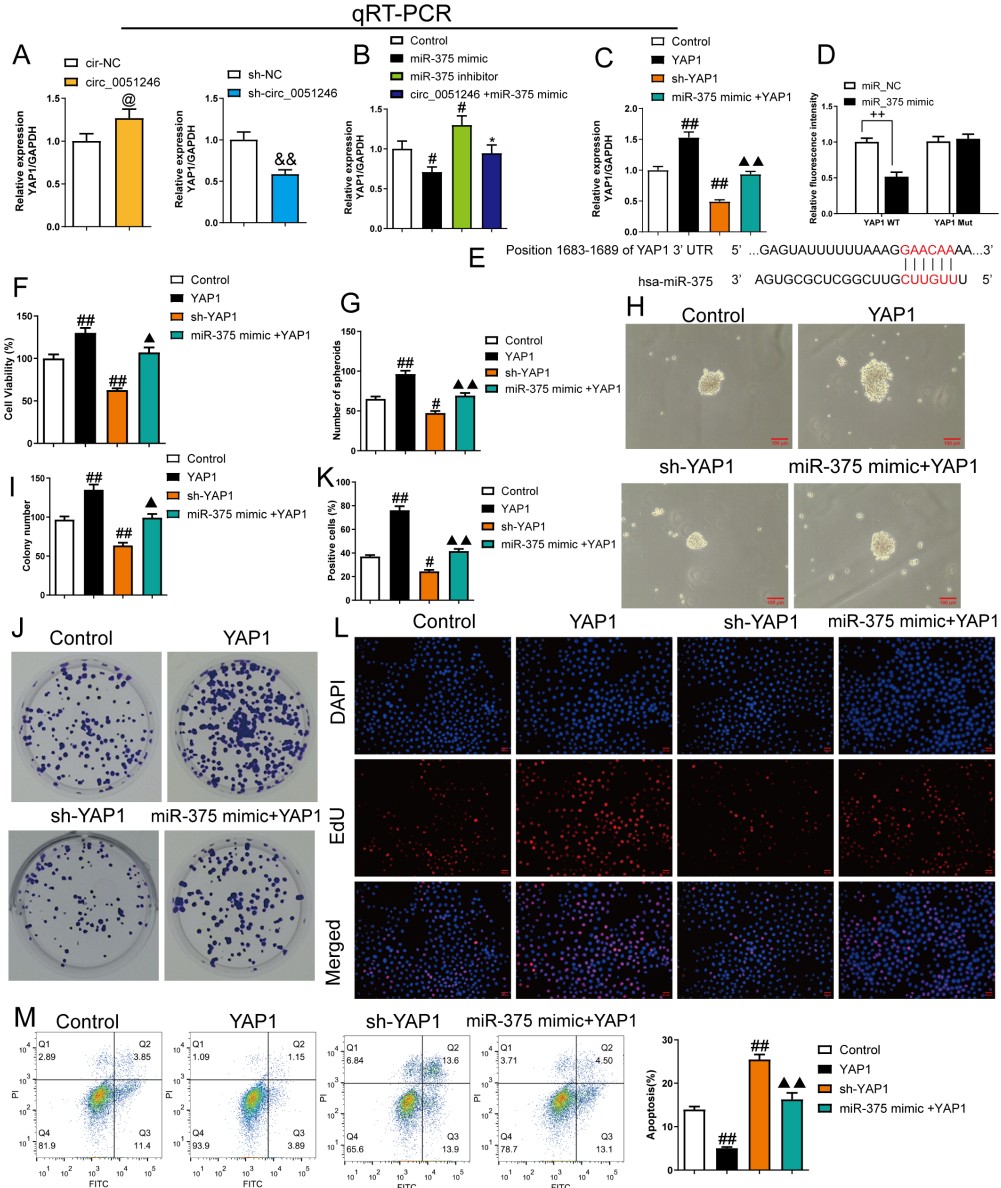

**Figure 7** **MiR-375 inhibited the effects of YAP1 on the proliferation, self-renewal, and apoptosis in SGC-7901 cancer stem cells (CSCs) ($n = 3$, mean ± SD).** (A) The YAP1 levels in the CSCs with circ_0051246 knockout or overexpression were measured using quantitative real-time PCR. [@]$P < 0.05$ vs. circ-NC group; [&&]$P < 0.01$ vs. sh-NC group. (B) YAP1 levels in the CSCs with miR-375 mimics, miR-375 inhibitors, or circ_0051246/miR-375 mimics treatment were measured using quantitative real-time PCR. [#]$P < 0.05$, [##]$P < 0.01$ vs. Control group; [*]$P < 0.05$, [**]$P < 0.01$ vs. miR-375 mimic group. (C) YAP1 mRNA levels in the CSCs with YAP1, YAP1 knockout, or miR-375/YAP1 treatment were measured using quantitative real-time PCR. [#]$P < 0.05$, [##]$P < 0.01$ vs. Control group; [▲▲]$P < 0.01$ vs. YAP1 group, mean ± SD. (D) The interaction between miR-375 and YAP1 was verified using the Dual-luciferase reporter assay. [++]$P < 0.01$ vs. miR-NC group. (E) Schematic Diagram of 3′-UTR binding sites of miR-375 and YAP1 genes. (F) The effects of miR-375 on the viability of CSCs with knockout and overexpression of YAP1 were measured using the CCK-8 assay. (G, H) The effects of miR-375 on the self-renewal ability of (continued on next page...)

**Figure 7 (…continued)**
CSCs with knockout and overexpression of YAP1 were observed using the sphere formation assay (scale bars = 100 mm). (I, J) The effects of miR-375 on colony-forming ability in CSCs with knockout and over-expression of YAP1 were observed using the colony formation assay. (K, L) The effects of miR-375 on the proliferation of the CSCs with YAP1 knockout and overexpression were measured using EdU cell prolif-eration imaging analysis (scale bars = 40 $\mu$m). (M) The effects of miR-375 on apoptosis in the CSCs with YAP1 knockout and overexpression were detected using flow cytometry. $^{\#}P < 0.05$, $^{\#\#}P < 0.01$ *vs.* Control group; $^{\blacktriangle\blacktriangle}P < 0.01$ *vs.* YAP1 group.

a significantly change in circ_0051246 levels (Fig. 9E). Additionally, silencing circ_0051246 and activating miR-375 markedly increased the miR-375 levels in the tumors ($P < 0.01$) (Fig. 9E). Furthermore, circ_0051246 knockdown and miR-375 activation inhibited YAP1 mRNA and protein expression (Figs. 9E–9G). Moreover, circ_0051246 knockdown and miR-375 activation inhibited Notch 1 and Jagged 1 protein levels in tumors ($P < 0.01$) (Figs. 9G and 9H). These results suggested that circ_0051246 inhibits miR-375 to promote the tumorigenicity of CSC cells *in vivo*.

## DISCUSSION

CSCs have become a crucial research topic in the field of tumor metastasis, as recent studies have indicated that they drive the proliferation and metastasis of cancer cells (*Batlle & Clevers, 2017*). Compared to differentiated and mature cancer cells, CSCs have stronger drug resistance to radiotherapy, chemotherapy and even biological therapy (*Nassar & Blanpain, 2016*). Owing to their strong invasiveness and drug resistance, CSCs are crucial cells that promote cancer metastasis and recurrence. Studies have reported a positive correlation between the expression of CSC markers in GC and invasion depth as well as lymph node involvement, and that CSC markers are related to prognosis (*Attia et al., 2019*). Currently, surface CSCs' markers in GC include CD44, EpCAM, CD24, and CD133 (*Nguyen et al., 2017*). CD44 and EpCAM are highly expressed in CSCs of GC and are both related to the invasion and infiltration of GC (*Chen et al., 2019*). Therefore, the combined detection of CD44 and EpCAM is typically used as an indicator for early diagnosis and prognosis of GC. CD44 and EpCAM double-positive cells were successfully screened from SGC-7901 cell lines, and the alignment was identified as CSCs.

Studies have found that miR-375 plays a dual role as both an oncogene and a tumor suppressor. MiR-375 levels are low in conditions such as GC (*Kang et al., 2018*), esophageal squamous carcinoma (*Wu et al., 2021*), and renal cancer (*Wang & Sun, 2018*). Conversely, it is upregulated in other cancers, including prostate cancer (*Wang et al., 2016*), breast cancer (*Tang et al., 2020b*) and nasopharyngeal cancer (*Jia-Yuan et al., 2020*). In this study, we found that miR-375 is associated with GC development and progression. Treatment with miR-375 mimics significantly decreased the proliferation, migration, and invasion of gastric CSCs, whereas cell apoptosis increased. Therefore, our findings suggest that miR-375 acts as a tumor suppressor gene in GC, which is consistent with previous studies (*Ni et al., 2021*; *Zhang et al., 2021*). We found that circ_0051246 serves as a miR-375 sponge that can rescue YAP1 expression, thus affecting its biological functions. In particular, circ_0051246

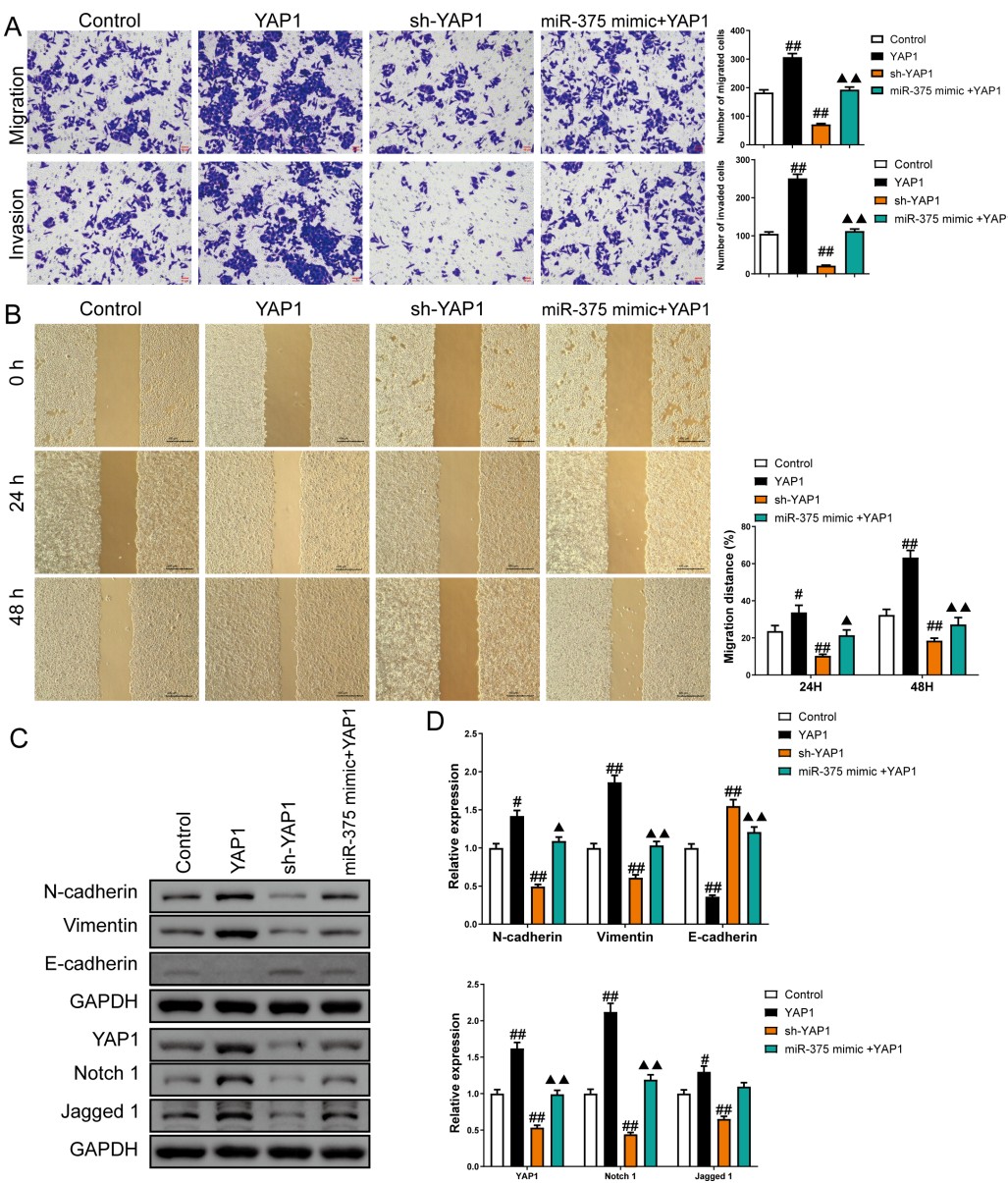

**Figure 8  MiR-375 inhibited the effects of YAP1 on cell migration, invasion, and related proteins in SGC-7901 cancer stem cells (CSCs) ($n = 3$, mean ± SD).** (A) The effects of miR-375 on migration and invasion in the CSCs with YAP1 overexpression were observed using the Transwell assay (scale bar = 40 μm). (B) The effects of miR-375 on migration in the CSCs with YAP1 overexpression were observed using the wound-healing assay (scale bar = 400 μm). (C) Western blot results indicating proteins associated with migration and YAP1 signaling pathway. (D) The effects of circ_0051246 on expression levels of proteins associated with migration and YAP1 signaling pathway in the CSCs with YAP1 overexpression were measured using Western blot. $^{\#}P < 0.05$, $^{\#\#}P < 0.01$ *vs.* Control group; $^{\blacktriangle}P < 0.05$, $^{\blacktriangle\blacktriangle}P < 0.01$ *vs.* YAP1 group.

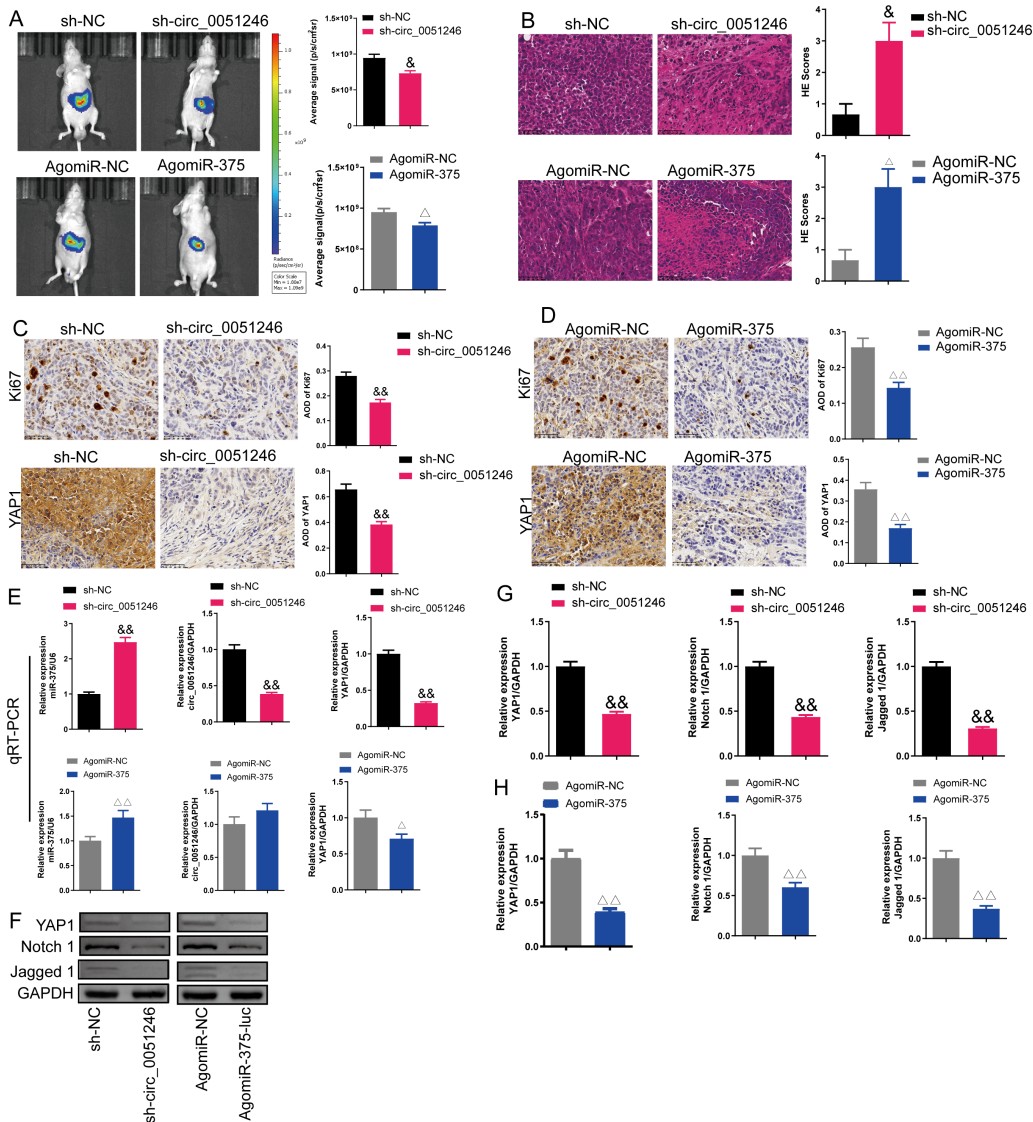

**Figure 9** **Effects of circ_0051246 and miR-375 on SGC-7901 cancer stem cells (CSCs) *in vivo*.** (A) The imaging system *in vivo* was used to observe the fluorescence signal intensity in tumor-bearing mice with silenced-circ 0051246 or activated-miR-375 ($n = 3$). (B) Histopathological observation of those tumors using HE staining ($n = 3$, ×400, scale bar = 50 μm). (C, D) The expression levels of Ki67 and YAP1 in tumor-bearing mice with silenced-circ 0051246 or activated-miR-375 were observed using immunohistochemistry ($n = 3$, ×400, scale bar = 50 μm). (E) The miR-375, circ_0051246, and YAP1 mRNA levels in tumor-bearing mice with silenced-circ 0051246 or activated-miR-375 were measured using quantitative real-time PCR ($n = 3$). (F) Western blot results showing YAP1, Notch 1, and Jagged 1 ($n = 3$). (G) The expression levels of YAP1, Notch 1, and Jagged 1 proteins in tumor-bearing mice whose tumors with silenced-circ 0051246 were measured using Western blot. (H) The expression levels of YAP1, Notch 1, and Jagged 1 proteins in tumor-bearing mice with activated-miR-375 were measured using Western blot. [&]$P < 0.05$, [&&]$P < 0.01$ *vs.* sh-NC group; [△]$P < 0.05$, [△△]$P < 0.01$ *vs.* AgomiR-NC group, mean ± SD.

functions as a miR-375 sponge and promotes the progression of CSCs by increasing YAP1 levels through miR-375 inhibition.

The Notch signaling pathway is composed of four major components: the Notch receptor, Notch ligand, CSL DNA-binding protein, and target genes. Within this pathway, there are four Notch receptors (Notch 1-4) and five ligands including delta-like 1, 3, 4, Jagged 1, and 2 (*Ranganathan, Weaver & Capobianco, 2011*). Several studies have demonstrated that the Notch signaling pathway serves as a vital regulator of CSCs, promotes self-renewal of CSCs, maintains the phenotype and plays a significant role in cancer cell progression (*Ibrahim et al., 2017*). Studies have found that the dysregulation of the Notch signaling pathway in human GC and the increased levels of Notch 1 and Jagged 1 are associated with the incidence of GC (*Demitrack & Samuelson, 2017*), but the mechanism of the Notch signaling pathway in GC remains unclear. Studies have reported that YAP1 promotes tumorigenesis and progression in hepatocellular carcinoma by upregulating Jagged 1 and activating the Notch pathway (*Ren et al., 2016*; *Tschaharganeh et al., 2013*). We showed that circ_0051246 inhibited miR-375 to upregulate YAP1 expression *in vitro* and *in vivo*, suggesting that circ_0051246 may be involved in the tumorigenesis of gastric CSCs.

In this study, circ_0051246 was found to modulate Bax and Bcl-2 protein levels in gastric CSCs. Bax plays a critical role in mitochondrial-mediated cell death by permeabilizing the outer mitochondrial membrane (*Spitz & Gavathiotis, 2022*). Conversely, the activation of Bcl-2 inhibits apoptosis. A previous study reported that Bcl-2 inhibitors target CSCs to overcome drug resistance (*Song et al., 2021*). In addition, downregulation of the E-cadherin gene can result in a decrease in its abundance on the cell membrane, thereby attenuating or eliminating cell–cell interactions. Loss of E-cadherin expression has been considered a major cause of the epithelial-mesenchymal transition (EMT). However, other studies have indicated that the loss of E-cadherin expression is a consequence rather than a cause of EMT (*Bure, Nemtsova & Zaletaev, 2019*). Our findings demonstrate that circ_0051246 overexpression inhibits apoptosis in gastric CSCs *via* miR-375, suggesting the potential of circ_0051246 knockdown to overcome drug resistance and suppress tumor metastasis and invasion.

We demonstrated that the knockdown of circ_0051246 inhibited the self-renewal, proliferation, migration, and invasion in CSCs of GC *via* the miR-375/YAP1 pathway. The Notch 1 pathway may also be involved in this process. Additionally, circ_0051246 indirectly regulates YAP1; however, its direct effect requires further experimental verification. A limitation of this study includes only observing the role of circ_0051246 in SGC-7901 CSCs, and experimental data *in vitro* and *in vivo* and clinical evidence are still needed to support our hypothesis. In addition, to explore the progression of GC, the effects of circ_0051246 and miR-375 on normal cells are worthy of attention. Overall, our results provide a scientific foundation for further investigations of interactions and mechanisms between circ_0051246 and miR-375 in GC.

## CONCLUSIONS

We found that overexpression of circ_0051246 or YAP1 in gastric CSCs promoted malignant development, increased protein levels of Ki67, PCNA, Bcl-2, YAP1, Notch 1, Jagged 1, N-cadherin, and vimentin, and inhibited the Bax and E-cadherin levels, whereas inhibition of circ_0051246 or YAP1 had the opposite effect. Additionally, miR-375 mimics caused a similar decrease in malignancy development as circ_0051246 knockdown; however, miR-375 inhibitors induced the opposite effect. Further analysis showed that circ_0051246 overexpression antagonized the effects of miR-375 mimics, and that miR-375 mimics antagonized the effects of YAP1 overexpression in CSCs. Finally, *in vivo* experiments demonstrated that circ_0051246 knockdown and miR-375 activation inhibited the tumorigenicity of gastric CSCs. In summary, our study shows that circ_0051246 sponges miR-375 to promote GC progression by increasing YAP1 expression, providing a scientific basis for further investigation into the occurrence and development of GC.

### Funding

This work was supported by the National Natural Science Foundation of China (No.: 82074209). The funders had no role in study design, data collection and analysis, decision to publish, or preparation of the manuscript.

### Grant Disclosures

The following grant information was disclosed by the authors:
National Natural Science Foundation of China: 82074209.

### Competing Interests

The authors declare there are no competing interests.

### Author Contributions

- Minghui Deng conceived and designed the experiments, authored or reviewed drafts of the article, and approved the final draft.
- Yefeng Xu performed the experiments, authored or reviewed drafts of the article, and approved the final draft.
- Yongwei Yao performed the experiments, prepared figures and/or tables, and approved the final draft.
- Yiqing Wang analyzed the data, prepared figures and/or tables, and approved the final draft.
- Qingying Yan analyzed the data, prepared figures and/or tables, and approved the final draft.
- Miao Cheng analyzed the data, prepared figures and/or tables, and approved the final draft.
- YunXia Liu conceived and designed the experiments, authored or reviewed drafts of the article, and approved the final draft.

## Animal Ethics

The following information was supplied relating to ethical approvals (*i.e.*, approving body and any reference numbers):

The Animal Experimentation Ethics Committee of Zhejiang Eyong Pharmaceutical Research and Development Center.

## Data Availability

The raw image of western blot and the raw data are available in the Supplementary Files.

## Supplemental Information

Supplemental information for this article can be found online at http://dx.doi.org/10.7717/peerj.16523#supplemental-information.

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
