# Peer review of "Circular RNA hsa_circ_0051246 acts as a microRNA-375 sponge to promote the progression of gastric cancer stem cells via YAP1"

_PeerJ, doi:10.7717/peerj.16523_

## Round 0.1 · original submission · Minor Revisions

Please carefully read the comments and provide your point-by-point responses.

**Language Note:** The review process has identified that the English language must be improved. PeerJ can provide language editing services - please contact us at copyediting@peerj.com for pricing (be sure to provide your manuscript number and title). Alternatively, you should make your own arrangements to improve the language quality and provide details in your response letter. – PeerJ Staff

Reviewer 1 ·

Basic reporting

This study investigated the effect of hsa_circ_0051246, microRNA-375, and YAP1 on gastric cancer stem cells, which is an interesting development in this field. Experiments were designed appropriately. However, there are still problems that need to be revised for potential publication. The points are summarized below:

Experimental design

1. Title: The "promote" should be changed to "promotes".

Abstract:
1. Abstract: The " Backgroud" should be changed to "Background".

2. "MiRNA participates in the progress of GC" should be " MiRNAs participate……".

3. Please indicate how many CSCs were injected during the construction of orthotopic xenograft tumor models.

4. The use of abbreviations in the Abstract is quite casual. For example, "CCK-8", "H&E", etc. appear only once in the Abstract and do not need to be abbreviated. Please use abbreviations in a standardized manner.

Introduction
1. CSC plays a crucial role in cancer, as the author acknowledges. However, its significance requires further explanation based on relevant literature. The scientific basis closely related to the occurrence, metastasis, or prognosis of GC should be explained by citing literature.

Materials and Methods
The author's methods are well-written, with a few minor revisions still needed.

1. In the cell transfection and grouping part, "purinomycin" should be "puromycin". Additionally, the dose and basic information of puromycin are lost.

2. "the interaction of miR-375 and circ_0051246, SGC-7901 GSCs……" is confusing.

3. How many cells are used for CCK-8 and Transwell assays?

4. In HE staining, "HE kti" should be "HE kit". Full text needs a thorough spell check.

Validity of the findings

Results
1. Results of Fig. 3A should specify the multiples compared to the sh-NC or circ-NC cells.
Discussion
1. "We proved that circ_0051246 could inhibit self-renewal, proliferation, migration, and invasion ……" contradicts the experimental results.

2. To explore the progression of GC, the effects of circ_0051246 and miR-375 in normal cells are also worth paying attention to. This should be discussed the limitations of the manuscript.

Additional comments

Figures
1. In Figure 7A, the group circ-NC names overlap.

Annotated reviews are not available for download in order to protect the identity of reviewers who chose to remain anonymous.

Reviewer 2 ·

Basic reporting

In this study, authors demonstrated that Circular RNA hsa_circ_0051246 acts as microRNA-375 sponge to promote the progression of gastric cancer stem cells via YAP1. MiR-375 mimics inhibited the degree of proliferation, self-renewal, migration, and invasion in gastric CSCs, while circ_0051246 overexpression antagonized the effects of miR-375 mimics treatment on gastric CSCs. Moreover, YAP1 overexpression promoted the levels of proliferation, self-renewal, migration, and invasion, and inhibited apoptosis degree, which was offset by miR-375 mimics addition. Besides, the orthotopic-xenograft model of gastric CSC proved the tumor inhibition effect of circ_0051246-silence and miR-375-activation. The findings provide a scientific basis for further investigation into the occurrence and development of gastric cancer. Despite the intriguing study, there are several points that require more careful examination and explanation.
Comments: 1)
There are many grammar and typing errors, detracting from the readability of the text. Such as: “Backgroud”, “Circular RNAs (circRNAs) as microRNAs (miRNAs) sponges in the cytoplasm inhibit miRNAs’ biological function in GC.” “H&E staning”, “Venna plots” “it had a high expression level in SGC-7901 and AGS cells (P<0.01) (figure 2A)”.
Comments: 2)
Title: “Circular RNA hsa_circ_0051246 as microRNA-375 sponge promote ...” should be changed to “Circular RNA hsa_circ_0051246 acts as microRNA-375 sponge to promote ...”
Comments: 3)
Abstract: the abstract is too long, and the content needs to be more focused. The background is muddled, the associations among Circular RNA hsa_circ_0051246-microRNA-375-YAP1 should be state accurate in the result section.

Experimental design

Comments: 4)
Introduction: in second paragraph, except for the critical role of circRNA in cancer, biological function of miRNA in cancers should also be introduced.
Comments: 5)
Materials and Methods: the cell counts for EdU staining is missing; the microscope information for HE and immunohistochemistry staining should be detailed; Why does the WB experiment have two internal reference? In Orthotopic xenograft tumor model, do the CSCs have fluorescence label, such as GFP, as the the intravital fluorescence imaging system of small animals was performed.

Validity of the findings

Comments: 6)
Results: Figures need to be sorted in the order they appear in the body of the text. The order for the results presentation of figures 3C-E and figures 3B, 3D-F should be interchanged. The subtitle “Interaction between circ_0051246 and miR-375 in SGC-7901 CSCs” is too general; the subtitle “MiR-375 inhibits the progression of SGC-7901 CSCs via YAP1”, how does miR-375 regulates YAP1, activates or inhibits?
Comments: 7)
Discussion: “and five ligands including Delta-like 1, 3, 4, Jagged 1, and Jagged 2”, maybe “and five ligands including Delta-like 1, 3, 4, Jagged 1 and 2”; the associations among hsa_circ_0051246-microRNA-375-YAP1, and their role in CSCs of GC should be state clearly in the second paragraph.
The conclusions need to be condensed.

Reviewer 3 ·

Basic reporting

In the present study, the authors have demonstrated that circ_0051246 RNA sponges miR-375 to promote the progression of gastric cancer via YAP1 in gastric cancer stem cells. The authors have provided sound experimental evidence by using silencing and overexpression approaches. Mechanistically, the study is detailed. The author has used both in vitro and in vivo experimental approaches to validate their findings. The MS is recommended for publication provided the authors address the following issues.
1. Please explain in Abstract where is YAP1 coming. “Many studies have confirmed that miR-375 could exert its functional effects by targeting several important oncogenes such as Yes-associated protein 1 (YAP1)”
2. In methods: cell colony formation section “CSCs (1000 cells) were incubated in per-well of the 6-well plates with RPMI 1640 culture medium (FBS is 30%) until cell count in every clone greater than 50 cells” The sentence do not make any sense. Please correct.
3. The authors should also work on language
4. Line 192-193 not clear
5. Line 204-206 not clear
6. mice (n=6) were divided into sh-NC, sh-circ_0051246, AgomiR-NC, and AgomiR-375 groups using the random table method. (line 219 to 220) Are these mice (n=6) per group? If yes, please mention.
7. If the mice experience a wright loss greater than 20%, if the tumor diameter is too large (weighting more than 5% of body weight), or if the tumor ruptures, euthanasia will be conducted using CO2 (line 221 to 222): The authors needs to work extensively in the language of methods section

Experimental design

8. Fig 2A: What is GES: the author should mention this in the corresponding results section as this is control
9. Fig 2C-2E: The author should state that these experiments were done in SGC-7901 CSCs in the results section as the clarity is missing

Validity of the findings

10. The authors have demonstrated that silencing of SGC-7901 CSCs with sh-circ 0051246, is protecting against disease pathology, but the authors have no where shown any rescue experiment by nullifying the effect of shRNA. Instead, they have overexpressed circ 0051246 and have found the deteriorating effect. What if the author uses an agonist for circ 0051246/or overexpression of same in sh-circ 0051246 cells to rescue the effect of silencing and to validate their findings. They can do this in one of the experiments.
11. In Figure 4F, the authors have directly jumped in the protein expression of YAP1, but they have not introduced anywhere in result section how is it relevant with respect to circ 0051246.
12. In Figure 5E-I, the author could also keep one more group of circ 0051246 to show that mir375 mimic is reducing the effect of circ 0051246. But since, it will be again a lot of experiments. They could just show doing one experiment e.g., no. of spheroid and putting the data in supplementary. This will further strengthen their finding.
13. Fig 5B, 7D: Dual-luciferase report assay: Will this give a fluorescence intensity or luminescence intensity? Please correct.

Additional comments

N/A

---

## Round 0.2 · accepted · Accept

The authors have addressed the comments from all reviewers and the manuscript may be acceptable for publication.

Reviewer 1 ·

Basic reporting

no comment

Experimental design

no comment

Validity of the findings

no comment

Reviewer 2 ·

Basic reporting

accept

Experimental design

accept

Validity of the findings

accept

Additional comments

accept

Reviewer 3 ·

Basic reporting

N/a

Experimental design

N/A

Validity of the findings

The authors have substantially improved the MS. It can now be published in present form